# Asian Sand Dust Particles Enhance the Development of *Aspergillus fumigatus* Biofilm on Nasal Epithelial Cells

**DOI:** 10.3390/ijms23063030

**Published:** 2022-03-11

**Authors:** Seung-Heon Shin, Mi-Kyung Ye, Dong-Won Lee, Mi-Hyun Chae

**Affiliations:** Department of Otolaryngology-Head and Neck Surgery, School of Medicine, Catholic University of Daegu, Daegu 42472, Korea; miky@cu.ac.kr (M.-K.Y.); neck@cu.ac.kr (D.-W.L.); leonen@hanmail.net (M.-H.C.)

**Keywords:** Asian sand dust, *Aspergillus fumigatus*, biofilm, epithelial cells, inflammatory mediator

## Abstract

Background: Asian sand dust (ASD) and *Aspergillus fumigatus* are known risk factors for airway mucosal inflammatory diseases. Bacterial and fungal biofilms commonly coexist in chronic rhinosinusitis and fungus balls. We evaluated the effects of ASD on the development of *A. fumigatus* biofilm formation on nasal epithelial cells. Methods: Primary nasal epithelial cells were cultured with *A. fumigatus* conidia with or without ASD for 72 h. The production of interleukin (IL)-6, IL-8, and transforming growth factor (TGF)-β1 from nasal epithelial cells was determined by the enzyme-linked immunosorbent assay. The effects of ASD on *A. fumigatus* biofilm formation were determined using crystal violet, concanavalin A, safranin staining, and confocal scanning laser microscopy. Results: ASD and *A. fumigatus* significantly enhanced the production of IL-6 and IL-8 from nasal epithelial cells. By coculturing *A. fumigatus* with ASD, the dry weight and safranin staining of the fungal biofilms significantly increased in a time-dependent manner. However, the increased level of crystal violet and concanavalin A stain decreased after 72 h of incubation. Conclusions: ASD and *A. fumigatus* induced the production of inflammatory chemical mediators from nasal epithelial cells. The exposure of *A. fumigatus* to ASD enhanced the formation of biofilms. The coexistence of ASD and *A. fumigatus* may increase the development of fungal biofilms and fungal inflammatory diseases in the sinonasal mucosa.

## 1. Introduction

*Aspergillus fumigatus* is ubiquitous in the environment and is commonly associated with a wide range of human airway diseases. *A. fumigatus* conidia enter the upper and lower respiratory tract by inhalation, but *A. fumigatus* is rarely pathogenic in healthy individuals due to the innate immune defense system of the airway mucosa, which includes macrophages, the mucociliary clearance system, and pattern recognition receptors in epithelial cells [1,2]. If the inhaled conidia are not removed, they can germinate to form hyphae. Filamentous growth and the dense accumulation of fungal hyphae are characteristics of fungus balls (FB). Sinus FB are the most common noninvasive fungal rhinosinusitis, and *A. fumigatus* is the main cause of FB [3]. *A. fumigatus* can form biofilms in multicellular components and the extracellular matrix [4]. Bacterial and fungal biofilms commonly coexist in the sinus cavity in sinus FB and in chronic rhinosinusitis (CRS) [5,6].

Asian sand dust (ASD) originates from the Gobi Desert and the Ocher Plateau. It contains various chemical and microbiological elements. The chemical materials that are produced by air pollutants in ASD are involved in neutrophilic inflammation, and microbiological materials contribute to the development of eosinophilic inflammation [7]. The inhaled ASD makes direct contact with respiratory epithelial cells and induces the production of proinflammatory and inflammatory mediators with inflammatory cell infiltration [8]. Around 53% to 71% of ASD consists of particulate matter measuring less than 10 μm (PM10), which influences pulmonary dysfunction. Particulate matter measuring less than 2.5 μm (PM2.5) affects nasal epithelial barrier dysfunction by decreasing tight junction proteins [9,10]. ASD also affects common cold and allergic rhinitis symptoms by enhancing the production of inflammatory chemical mediators from respiratory epithelial cells, which aggravate both neutrophilic and eosinophilic inflammation [8,11].

Global warming, increased air pollution, longer life expectancy, the increased use of antibiotics, and increased allergen exposure could influence the development of FB and CRS; however, the etiology and pathogenesis of FB and CRS are not clearly understood [12,13]. The increase in the concentration of ASD air pollution is associated with an increase in the incidence and morbidity of respiratory diseases. The microbial and chemical components of ASD can induce mucosal inflammation after contact with airway epithelial cells. Fungal biofilms exist on the sinus mucosal surfaces in FB and CRS and result in increased mucosal inflammation [5,6]. ASD and *A. fumigatus* are risk factors for FB and CRS. The presence of ASD decreases host immune defense and increases cell susceptibility to bacterial infection and biofilm formation in the airway mucosa [14]. ASD particles may provide a favorable surface for fungal attachment and biofilm growth. In this study, we aimed to determine whether ASD particles could affect *A. fumigatus*-induced inflammatory chemical mediator production from nasal epithelial cells and the development of fungal biofilms on nasal epithelial cells.

## 2. Results

### 2.1. Cytotoxic Effects of ASD on Nasal Epithelial Cells

Primary nasal epithelial cells were treated with 0 to 500 μg/mL of ASD for 72 h. Cell proliferation was found to be significantly decreased at ASD concentrations exceeding 200 μg/mL. More than 30% of the absorbances decreased with 500 μg/mL ASD at each time point, and more than 20% decreased with 200 μg/mL ASD after 72 h of incubation (Figure 1). Therefore, we used less than 100 μg/mL of ASD for the subsequent experiments.

### 2.2. Effects of ASD and A. fumigatus on the Production of Inflammatory Chemical Mediators from Nasal Epithelial Cells

The levels of Interleukin (IL)-6, IL-8, and transforming growth factor (TGF)-β1 in the supernatants were measured after 24 h, 48 h, and 72 h of treatment with ASD and *A. fumigatus*. The levels of IL-6 and IL-8 were found to be significantly increased by treatment with 50 and 100 μg/mL of ASD. *A. fumigatus* was found to significantly enhance the production of IL-6 (at 24 h and 48 h) and IL-8 (at each time point), but not the TGF-β1 from nasal epithelial cells. However, ASD and *A. fumigatus* did not affect the production of TGF-β1 from the nasal epithelial cells. The coculture with ASD and *A. fumigatus* did not enhance or even suppress (IL-6 at 24 and 48 h, IL-8 at 48 h) the production of inflammatory mediators from nasal epithelial cells (Figure 2).

### 2.3. Effects of ASD on the Development of Biofilms on Nasal Epithelial Cells

When *A. fumigatus* was cultured with nasal epithelial cells, the biofilm dry weight was 1.6 ± 0.3 mg at 24 h, 2.5 ± 0.3 mg at 48 h, and 2.6 ± 0.5 mg at 72 h. The coculture of *A. fumigatus* with ASD in the nasal epithelial cells increased the biofilm dry weight in a concentration- and time-dependent manner (with 50 μg/mL of ASD: 2.2 ± 0.4 mg, 3.0 ± 0.3 mg, and 3.2 ± 0.4 mg at 24, 48, and 72 h, respectively; with 100 μg/mL of ASD: 2.3 ± 0.4 mg, 3.6 ± 0.5 mg, and 3.7 ± 0.4 mg at 24, 48, and 72 h, respectively). The presence of ASD significantly increased the dry weight of *A. fumigatus* biofilm in comparison to *A. fumigatus* that had been cultured with or without nasal epithelial cells (Figure 3A).

Crystal violet was used to determine extracellular matrix (ECM) and the cell viability and to quantify the biofilm biomass [15]. When *A. fumigatus* was cultured with nasal epithelial cells, the quantity of ECM increased significantly at each time point in comparison to the wells, which contained *A. fumigatus* conidia cultured without nasal epithelial cells. However, the coculture of ASD with *A. fumigatus* did not have crystal violet staining intensities that were significantly higher compared to those obtained with *A. fumigatus* incubated without ASD. At 72 h, in the *A. fumigatus* cultured with ASD in nasal epithelial cells, the crystal violet staining intensities decreased (Figure 3B).

Safranin is a non-toxic staining method that is used to determine relative levels of biofilm formation [16]. The safranin staining densities were significantly increased in a time-dependent manner. When *A. fumigatus* was cultured with ASD in nasal epithelial cells, the safranin staining densities were significantly increased at each time point compared to the *A. fumigatus* that had been cultured with or without nasal epithelial cells (Figure 3C).

Concanavalin A stains the cell wall structures that have high concentrations of carbohydrates [17]. The biological activity of *A. fumigatus* lectin expression increased significantly with the administration of 50 μg/mL and 100 μg/mL of ASD (1.5 ± 0.3 and 1.8 ± 0.3, respectively) at 24 h and with 50 μg/mL of ASD (4.0 ± 0.6) at 48 h in compared to when *A. fumigatus* was cultured without ASD (0.8 ± 0.2 at 24 h and 2.8 ± 1.1 at 48 h). However, at 72 h, the biological activity was significantly decreased by the coculture of *A. fumigatus* and ASD in nasal epithelial cells (2.0 ± 0.3 with 50 μg/mL of ASD and 1.9 ± 0.5 with 100 μg/mL of ASD) compared to the optical density at 48 h (Figure 3D).

### 2.4. Confocal Laser Scanning Microscopic Findings

To analyze the structure of the *A. fumigatus* biofilms, the ECM of the fungal cell wall was stained with concanavalin A and FUN-1, which stain the cytoplasm of metabolically active cells. *A. fumigatus* and nasal epithelial cells cocultures showed fungal growth with a hyphae network after 24 h (Figure 4A). The number of orange-red fluorescing sites (metabolically active fungal conidia and hyphae) and intense green sites (polysaccharides and ECM present between fungal hyphae) were increased in a time dependent manner (Figure 4B,C). The coculture of *A. fumigatus* and ASD in nasal epithelial cells significantly enhanced the intensity of the concanavalin A (Figure 4D) and FUN-1 (Figure 4E) at 48 h and 72 h of incubation.

## 3. Discussion

ASD originates from arid and semi-arid regions and their components can change during long-range transport, meaning that ASD mixes with various chemical and microbiological materials, such as sulfate, nitrate, pollen, bacteria, and fungi [7,8]. The microbial material in ASD enhances allergic and infectious airway inflammation [7]. The composition of the ASD mixture, which contains organic, inorganic, and microbial components, depends on the location where it is collected, the sources of emission near the collection site, and the season in which the dust is sampled. In this study, we autoclaved ASD to eliminate the microbial effect on nasal epithelial cells and A. fumigatus biofilm formation. The median size of the ASD particles was 6.06 μm, and the most abundant component was SiO (52.1%) (Appendix A). The ASD particles induced inflammatory chemical mediator production from the nasal epithelial cells, and the coculture of the A. fumigatus conidia and ASD particles enhanced biofilm formation on the nasal epithelial cells.

The respiratory epithelial cells act as a first defense system against inhaled environmental pathogens by providing a physical barrier and producing various chemical mediators. They also actively participate in the inflammatory process of the airway mucosa through the production of lipid mediators, cytokines, chemokines, and growth factors. ASD particles have been reported to increase the production of IL-6 and IL-8 from bronchial epithelial cells [18]. In this study, ASD particles also significantly enhanced the production of IL-6 and IL-8 in a dose-dependent manner. Additionally, the viability of the nasal epithelial cells tended to increase at higher ASD concentrations. The exposure time to ASD did not influence the production of these inflammatory chemical mediators from nasal epithelial cells. IL-6 and IL-8 are proinflammatory cytokines that participate in acute inflammatory responses, the recruitment of neutrophils, and the induction of mucin production [19]. ASD could enhance mucus production via the TLR2-dependent ERK2 and p38 signaling pathway in airway epithelial cells [20]. Although, *A. fumigatus* enhanced the production of IL-6 and IL-8 from nasal epithelial cells, the coculture of ASD with *A. fumigatus* did not enhance the production of IL-6 and IL-8 from nasal epithelial cells. In fact, the IL-6 and IL-8 levels decreased in the coculture comprising *A. fumigatus* and ASD in compared to ASD or *A. fumigatus* alone. Although ASD and *A. fumigatus* activate nasal epithelial cells, they also have cellular cytotoxicity. When nasal epithelial cells were cultured with *A. fumigatus*, the cells underwent morphological changes, including shrinkage and cellular detachment, after 48 h of treatment. These morphological changes may be associated with the proteolytic degradation of epithelial cells with decreased cell survival (Appendix A). In the presence of ASD, the protease activity of *A. fumigatus* was significantly increased at 72 h (*A. fumigatus* alone: 173.6 ± 90.4 relative fluorescence units (RFU); *A. fumigatus* and epithelial cells: 1834.7 ± 228.6 RFU; coculture with 50 μg/mL of ASD: 1970.5 ± 303.9 RFU; coculture with 100 μg/mL of ASD: 2164.7 ± 252.4 RFU) (Appendix A). *A. fumigatus* conidia directly interfere with the physical barrier via cellular detachment, which leads to the loss of cilia. *A. fumigatus* can also produce several epithelial toxins, such as tryptoquivaline, questin, and trypacidin and can induce an impaired mucociliary clearance mechanism [21]. The cellular toxicity of ASD and *A. fumigatus* may interfere with the innate mucosal defense system and enhance the susceptibility to invasion and the growth of pathogens in the sinonasal mucosa.

Several hundred conidia enter the airway each day, and less than 1% of airborne mold spores represent *A. fumigatus* [22]. With persistent contact conidia can germinate and produce mycelium, biofilm is created via multicellular community formation and interactions with polysaccharides in the ECM [23]. Fungal biofilms hamper the influx of nutrients and the disposal of waste products and reduce drug susceptibility. The in vitro culture of *A. fumigatus* conidia showed that conidia can germinate and produce mycelium. Interactions between polysaccharides in the ECM and fungal hyphae is an important component in the development of fungal biofilms. The increased dry weight of *A. fumigatus* and increased safranin staining, which indicates polysaccharides in the ECM, represent the increased formation of *A. fumigatus* biofilm on nasal epithelial cells. The dry weight and safranin staining intensity were further increased when the *A. fumigatus* was cultured with ASD. Increased biofilm formation in the presence of ASD may be associated with increased fungal growth and the increased metabolic activity of *A. fumigatus*. ASD provides a surface for fungal attachment and growth. The metallic components of ASD, such as iron, magnesium, and manganese, affect the production and metabolism of the fungal biomass [24]. The metallic components also increase the generation of pneumococcal biofilms and the colonization in middle ear epithelial cells [14]. Bacterial and fungal biofilms coexist in CRS patients, and these organisms synergistically interact to contribute to the development and survival of biofilms [6]. Although it is not possible to draw a clear conclusion, ASD may aggravate the formation of not only bacterial biofilms but also fungal biofilm in respiratory tract inflammatory conditions.

Due to the cytotoxicity of ASD on nasal epithelial cells, we used less than 100 μg/mL of ASD in this study. *A. fumigatus* also decreases the epithelial cell viability via morphological changes. Therefore, the coculture of ASD and *A. fumigatus* significantly influenced the viability of the epithelial cells. Crystal violet was used to determine cell viability and to quantify cell migration. When *A. fumigatus* was cultured with or without nasal epithelial cells, the optical density of the crystal violet increased in a time-dependent manner. However, in the coculture of *A. fumigatus* with ASD, the crystal violet levels were decreased at 72 h. Concanavalin A stains metabolically active fungi. The optical density of concanavalin A was also decreased at 72 h in the coculture of *A. fumigatus* with ASD. Although we cannot precisely and exactly explain this phenomenon, the decreased fungal viability and metabolic activity may be associated with a decreased production of inflammatory mediators or growth factors from nasal epithelial cells or with a decreased metallic concentration of ASD, which influences the growth and development of fungal biofilms in media with increased culture times.

There are some limitations in explaining the pathophysiologic phenomenon occurring in the sinonasal mucosa. First, autoclaved ASD were used; therefore, we cannot suggest the role of microbial components for the development of A. fumigatus biofilm. Second, ASD and *A. fumigatus* conidia influence the survival of nasal epithelial cells; however, in the sinonasal mucosa, epithelial cells can regenerate or change their characteristics via environmental stimuli, which may alter the development of biofilms. Third, 1 × 10^5^/mL of conidia was used to develop the biofilm; however, we do not know the concentration or the number of conidia in the sinonasal mucosa.

## 4. Materials and Methods

### 4.1. Preparation of ASD and Aspergillus fumigatus Spores

ASD was generously provided by You-Jin Hwang (Department of Life Science, College of Bionano, Gachon University, Incheon, Korea) and was collected from air dust using a high-volume air sampler (HV-500F, Sibata, Japan) during an ASD warning period in Incheon, South Korea. After the dust was collected, the filter paper was washed with 10 mL of phosphate-buffered solution (PBS). The fluid was filtered, and the particulate matter was collected and then centrifuged. The collected ASD was autoclaved at 121 °C for 15 min and stored in a freezer at −20 °C.

*A. fumigatus* was generously provided by Hun-Suck Seo (Department of Clinical Pathology, School of Medicine, Daegu Catholic University, Daegu, Korea). Conidia were collected by washing the plate surface with 5 mL of PBS plus 0.05% Tween 20, and 2 × 10^7^/mL conidia suspensions were placed at 45 °C to dry, after which they were stored at −80 °C until use.

### 4.2. Primary Nasal Epithelial Cell Culture and Cell Viability Assays

Primary nasal epithelial cells were isolated from the inferior turbinate of 10 subjects (6 men and 4 women; 44.7 ± 13.6 years old) during septal surgery. Subjects were excluded if they had active inflammation, allergies, or had received antibiotics, antihistamines, or other medications at least 4 weeks preoperatively. Allergy status was determined using the skin prick test. The Institutional Review Board of Daegu Catholic University Medical Center approved this study, and all subjects signed a consent form that outlined the study objectives.

Specimens were placed in Ham’s F-12 medium supplemented with 100 IU of penicillin, 100 µg/mL streptomycin, and 2 µg/mL amphotericin B, and were then transported to the laboratory. Primary nasal epithelial cells were isolated using 0.1% dispase (Roche Diagnostics, Mannheim, Germany), as previously described [25]. Cell suspensions (1 × 10^6^ cells/mL) were plated in 6-well culture plates (140675, Thermo Fisher Scientific, Waltham, MA, USA) and grown to 80–90% confluence on culture plates in a 5% CO_2_ humidified incubator at 37 °C.

To determine the cytotoxic effects of ASD, 200 μL of nasal epithelial cells (1 × 10^5^ cells/mL) were incubated in 96-well tissue culture plates (167008, Thermo Fisher Scientific) with 0, 10, 100, 200, or 500 μg/mL of ASD for 72 h. Cell cytotoxicity was determined using a CellTiter-96^®^ Aqueous One Solution Cell Proliferation Assay kit (Promega, Madison, WI, USA). For this assay, tetrazolium compound and phenazine ethosulfate were added to each well, and the plates were incubated for 4 h at 37 °C in a 5% CO_2_ chamber. Color intensities were assessed using a microplate reader at wavelength of 490 nm.

### 4.3. Analysis of Inflammatory Chemical Mediator Protein Production

When the nasal epithelial cells reached 80–90% confluence, the cells were treated with *A. fumigatus* conidia (1 × 10^5^/mL) for 2 h and then washed to remove the non-adherent fungi. Then, the nasal epithelial cells were cultured in media, including 50 µg/mL or 100 µg/mL ASD for 24 h, 48 h, and 72 h. The IL-6, IL-8, and TGF-β1 levels in the supernatants were quantified using an enzyme-linked immunosorbent assay (ELISA) kit (R&D Systems, Minneapolis, MN, USA). The absorbance at 450 nm was determined using an ELISA reader (BMG Labtech, Ortenaukreis, Germany).

### 4.4. Biofilm Dry Weight

Cultured materials were collected by scraping after 24 h, 48 h, and 72 h. The collected materials were filtered through a 0.45-μm cellulose nitrate membrane (Sartorius, Göttingen, Germany) and dried to a constant weight at 40 °C.

### 4.5. Crystal Violet Staining

At each time interval, after removing the culture media, the culture plate was air-dried, and 100 μL of 0.5% (*w*/*v*) crystal violet solution was added to the 96 well plate (167008, Thermo Fisher Scientific) for 5 min. The solution was carefully removed and washed with PBS to remove any excess stain. The biofilm was destained by adding 100 μL of 95% ethanol to each well for 1 min. The ethanol was transferred to a clean 96-well culture plate, and the optical density was measured at 570 nm with a spectrophotometer (FLUO Star Optima, BMG Labtech, Oldenburg, Germany). The absorbance value represents the quantities of the hyphae and the extracellular polymeric materials.

### 4.6. Safranin and Concanavalin A Staining

At each time interval, after removing the culture media, the polysaccharide structure of the extracellular matrix was stained with 50 μL of safranin solution for 5 min in a 96-well culture plate (167008, Thermo Fisher Scientific). After careful washing with PBS, the optical density was measured with a spectrophotometer (BMG Labtech) at 492 nm.

To stain the extracellular fungal elements, 100 μL of Alexa Fluor 488 conjugated to succinylated concanavalin A (Invitrogen, Carlsbad, CA, USA) was added to the 96-well culture plate (167008, Thermo Fisher Scientific) for 45 min. After being washed with PBS, the fluorescence intensity was measured at excitation and emission wavelengths of 485 nm and 520 nm, respectively, with a spectrophotometer (BMG LabTech).

### 4.7. Confocal Laser Scanning Microscopy

At each time interval, the 4-well Lab-Tek chamber slide (177399, Thermo Fisher Scientific) was stained with 100 μM of FUN-1 and 100 μg/mL of concanavalin A (Invitrogen) for 45 min in the dark. After being washed with PBS, image capture and analyses were performed using the Nikon A1 confocal microscope (Tokyo, Japan). The green fluorescent intracellular staining (concanavalin A) represents cell wall-like polysaccharides, and the orange-red fluorescent stain (FUN-1 cell stain) is localized in the cytoplasm of metabolically active cells. Yellow areas represent dual staining.

### 4.8. Statistical Analysis

All of the experiments were performed on at least five independent individuals, and every experiment was performed in duplicate. They revealed comparable results. The results are expressed as the mean ± standard deviation. Statistical significance was determined using Student’s *t*-test to make comparisons between the two groups, and data among several groups were analyzed by one-way analysis of variance followed by Tukey’s test. Statistical analyses were performed in SPSS ver. 25.0 (IBM Corp., Armonk, NY, USA). A *p*-value of 0.05 or less was considered statistically significant.

## 5. Conclusions

In the sinonsal mucosa, inhaled *A. fumigatus* conidia adhere to the epithelial surface, then colonize cells, form hyphae, and produce an ECM to form a biofilm. The present study shows that ASD particles and *A. fumigatus* enhance the production of inflammatory chemical mediators and that a coculture of ASD and *A. fumigatus* enhanced the development of fungal biofilms in primary nasal epithelial cells. During exposure to ASD, the copresence of inhaled ubiquitous A. fumigatus conidia in sinonasal mucosa affects innate immune defense system and aggravates inflammatory responses with increased fungal biofilm formation. Altogether, the coexistence of ASD particles and *A. fumigatus* increases the risk of the development of fungal biofilm and fungal inflammatory diseases in the sinonasal mucosa.

## Figures and Tables

**Figure 1 ijms-23-03030-f001:**
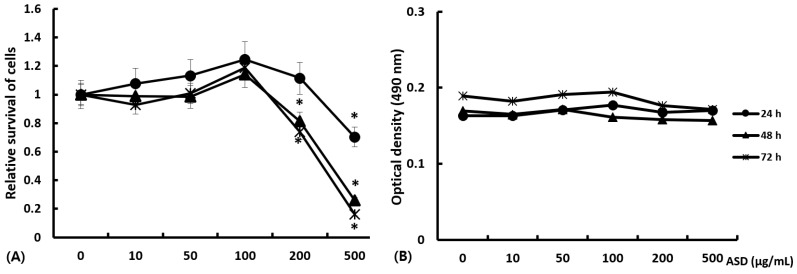
Cell viability effect of Asian sand dust (ASD) on primary nasal epithelial cells. (**A**) shows relative survival of nasal epithelial cells treated with 0 to 500 μg/mL of ASD for 72 h. Cell survival was significantly decreased above ASD concentrations of 200 μg/mL. Stimulation time did not influence the survival of nasal epithelial cells. (**B**) shows absolute optical density of different ASD concentrations without cells. *: *p* < 0.05 compared to without ASD, *k* = 10.

**Figure 2 ijms-23-03030-f002:**
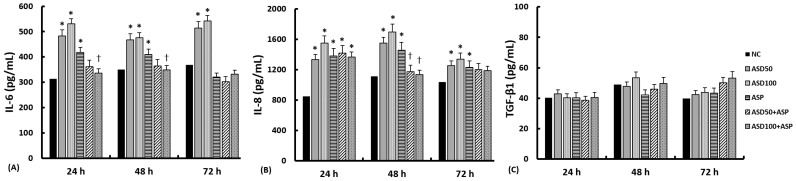
Effect of *Aspergillus fumigatus* (ASP) and Asian sand dust (ASD) on chemical mediator production in primary nasal epithelial cells. Doses of 50 μg/mL ASD (ASD50) and 100 μg/mL ASD (ASD100) enhanced interleukin (IL)-6 (**A**) and IL-8 (**B**) production at each time point. ASP also enhanced IL-8 at every time point and IL-6 at 24 h and 48 h. The coculture of ASP and ASD did not enhance IL-6 and IL-8 production. Transforming growth factor (TGF)-β1 (**C**) production was not influenced by stimulation with ASP and/or ASD. NC: negative control, *: *p* < 0.05 compared to NC, *k* = 14.

**Figure 3 ijms-23-03030-f003:**
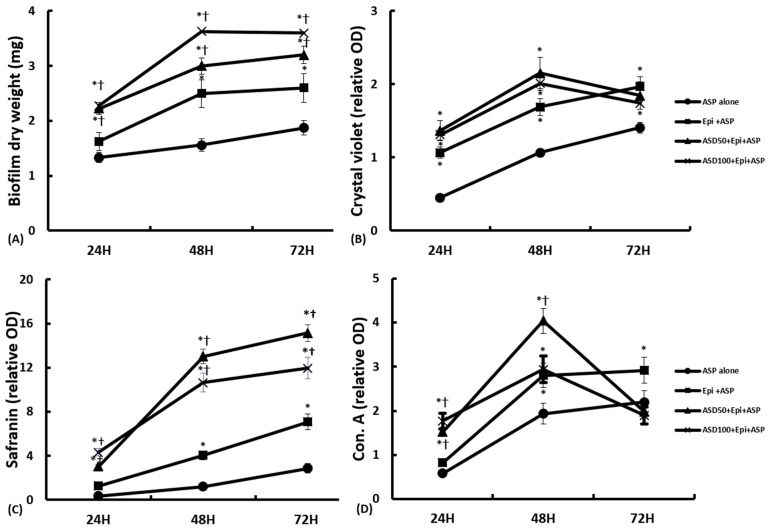
Quantification of *Aspergillus fumigatus* (ASP) biofilm in coculture of Asian sand dust (ASD) in primary nasal epithelial cells. The biofilm dry weight (**A**) and safranin staining (**C**) intensity significantly increased in a time-dependent manner, and the ASP and ASD coculture in nasal epithelial cells (ASD+Epi+ASP) significantly increased the biofilm dry weight and safranin intensity compared to ASP cultured with nasal epithelial cells (Epi+ASP) or without nasal epithelial cells (ASP alone). When the ASP was cocultured with 50 μg/mL of ASD (ASD50) or 100 μg/mL of ASD (ASD100), the crystal violet (**B**) and concanavalin A (Con. A) staining (**D**) intensities were increased at 24 h and 48 h of incubation. However, their staining intensities were decreased at 72 h. *: *p* < 0.05 compared to ASP alone, †: *p* < 0.05 compared to Epi+ASP, *k* = 14.

**Figure 4 ijms-23-03030-f004:**
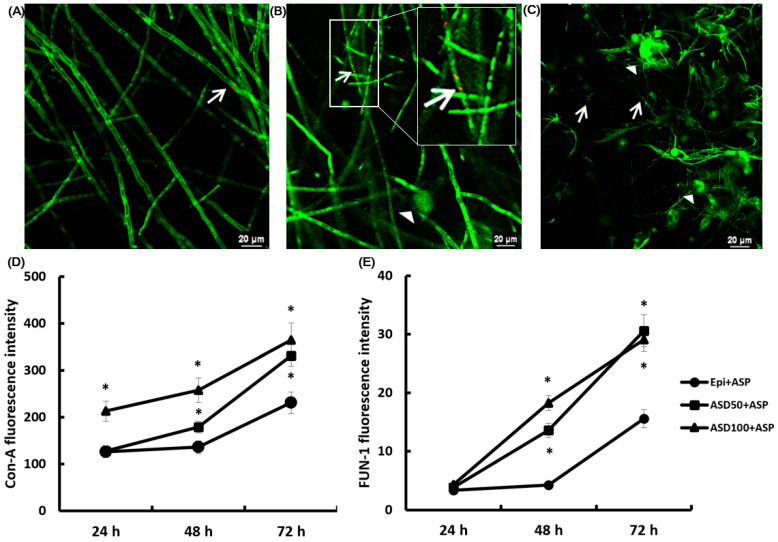
Confocal scanning laser microscopy findings of *Aspergillus fumigatus* (ASP) treated with Asian sand dust (ASD). (**A**–**C**) show the representative confocal scanning laser microscopic findings. *A. fumigatus* with nasal epithelial cells (Epi+ASP) at 24 h (**A**), 48 h (**B**), and 72 h ((**C**) 60 × magnification). The arrowhead indicates concanavalin A (Con-A) stain binding at the fungal cell wall’s polysaccharide and extracellular matrix, and the arrow indicates the FUN-1 stain binding to metabolically active fungi. When ASP was incubated with primary nasal epithelial cells, the fluorescent intensity of Con-A (**D**) and FUN-1 (**E**) increased in a time-dependent manner. The coculture of ASP and 50 μg/mL of ASD (ASD50) or 100 μg/mL of ASD (ASD100) significantly enhanced the Con-A and FUN-1 intensity at 48 h and 72 h. *: *p* < 0.05 compared to Epi+ASP, *k* = 10.

## Data Availability

Data supporting this study can be obtained by contacting the corresponding author.

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
