# Peer review of "Asian Sand Dust Particles Enhance the Development of Aspergillus fumigatus Biofilm on Nasal Epithelial Cells"

_ijms, 2022, doi:10.3390/ijms23063030_

Round 1

Reviewer 1 Report

The manuscript aims to characterize the toxic effects of Asian dust (ASD) and Aspergillus fumigatus (ASP) on human primary nasal epithelial cells. The authors assessed cell viability, the levels of IL6, IL8 and TGF beta in the supernatant as well as the biofilm formation/ fungal extracellular matrix.

The authors concluded on the inflammatory potential of ASD and ASP and their ability to favor biofilm formation in vitro.

The manuscript is well organized and reads well however there are numbers of issues that the manuscript suffers of.

  • It is unclear for the reviewer which type of plate was used for culture and viability test.
  • The number of the primary cell used in the exposure is missing.
  • Is there any characterization of the size or the shape of particles in ASD?
  • The authors reported that they performed the experiments at least 5 times and thus by reporting n=5 means 5 experiments. It little bit strange to run 5 in vitro experiments without replicate. Any specific reason for that? The reviewer wonders if it is the number of replicates and not the number of independent experiments the authors meant.
  • A control of the testing materials alone is missing. As such, the conclusions based on OD measurements need to be revised in the light of background levels.
  • Another critical issue is that authors reported in figure S2 protease activity, but there no description of how that was measured in the method section.
  • The scale bars in fig 2 A-C are missing
  • The reviewer wonders if the authors can present the limitations of the approach adopted.
  • Lines 102-105: the meaning of the sentence is not clear for the reviewer
  • Line 151: autoclaving eliminate growth but not microbiological effects
  • Line 184-185: Any reference for that.

Round II

  1. Regarding the plates (6 wells and 96 wells), can the authors provide the product reference and producer? The reviewer considers this info as critical for other researchers that will try to repeat the experiment.

  1. The authors reported that they performed the experiments at least 5 times and thus by reporting n=5 means 5 experiments. It was a little bit strange to run 5 in vitro experiments without replicating. Any specific reason for that? The reviewer wonders if it is the number of replicates and not the number of independent experiments the authors meant.

Answer) To clarify, authors described as ‘All the experiments were performed on at least five independent individuals and every experiment was performed in duplicate.’ In line 309-310.

The authors run at least 5 experiments and evry experiment in duplicate. The n for such setup will be 10. The reviewer is not convinced by the answer provided and will appreciate if the authors can give an explanation for the n values in fig 1 (n=5), fig 2 and 3 (n=7), fig 4 (n=5.1), figS2 and figS4 (n=5).

  1. Control of the testing materials alone is missing. As such, the conclusions based on OD measurements need to be revised in the light of background levels.

Answer) Fig 3B to D, we expressed the intensity of special staining as relative OD, which means absolute OD of backgrounds were converted into 1 and the measured ODs were expressed as relative OD compare with backgrounds.

What is the OD (at 490nm) of the ASD alone at different concentrations reported in fig 1?

Round III

  1. The authors reported that they performed the experiments at least 5 times and thus by reporting n=5 means 5 experiments. It was a little bit strange to run 5 in vitro experiments without replicating. Any specific reason for that? The reviewer wonders if it is the number of replicates and not the number of independent experiments the authors meant.

Answer) To clarify, authors described as ‘All the experiments were performed on at least five independent individuals and every experiment was performed in duplicate.’ In line 309-310.

The authors run at least 5 experiments and evry experiment in duplicate. The n for such setup will be 10. The reviewer is not convinced by the answer provided and will appreciate if the authors can give an explanation for the n values in fig 1 (n=5), fig 2 and 3 (n=7), fig 4 (n=5.1), figS2 and figS4 (n=5).

New comments: The reviewer cannot see any additional update on the number of treatments reported in different figures

  1. Control of the testing materials alone is missing. As such, the conclusions based on OD measurements need to be revised in the light of background levels.

Answer) Fig 3B to D, we expressed the intensity of special staining as relative OD, which means absolute OD of backgrounds were converted into 1 and the measured ODs were expressed as relative OD compare with backgrounds.

What is the OD (at 490nm) of the ASD alone at different concentrations reported in fig 1?

New comments: This question remains unanswered

Author Response

Last time I sent the revision file with responses to the reviewers, however it seems that you did not check it, so I am attaching the file again.

Reviewer 2 Report

After authors first revision, I still have two concerns:

Figure 1 – The authors should mention an explanation why the cell viability increases on 100 ug/ml

Figure 4 – I found very hard to see the results for the pictures. Can the authors consider making a picture amplification around red dots (as an example)? It would help readers to distinguish better the result.

Author Response

After authors first revision, I still have two concerns:

Figure 1 – The authors should mention an explanation why the cell viability increases on 100 ug/ml

Answer) To explain increased viability of epithelial cells at 100 ug/ml, ‘And the nasal epithelial cells viability tended to increase at higher concentration of ASD.’ Was added in line 173-174.

Figure 4 – I found very hard to see the results for the pictures. Can the authors consider making a picture amplification around red dots (as an example)? It would help readers to distinguish better the result.

Answer) Fig 4-B was changed as recommended.

Thank you.

Round 2

Round 4

  1. The authors reported that they performed the experiments at least 5 times and thus by reporting n=5 means 5 experiments. It was a little bit strange to run 5 in vitro experiments without replicating. Any specific reason for that? The reviewer wonders if it is the number of replicates and not the number of independent experiments the authors meant.

Answer) To clarify, authors described as ‘All the experiments were performed on at least five independent individuals and every experiment was performed in duplicate.’ In line 309-310.

The authors run at least 5 experiments and evry experiment in duplicate. The n for such setup will be 10. The reviewer is not convinced by the answer provided and will appreciate if the authors can give an explanation for the n values in fig 1 (n=5), fig 2 and 3 (n=7), fig 4 (n=5.1), figS2 and figS4 (n=5).

New comments: The reviewer cannot see any additional update on the number of treatments reported in different figures

Answer) In this study, n means the number of subjects which we isolated nasal epithelial cells for this study. So the total experiment numbers different from n. If we need to change the number of experiments, I will do as recommended.

New comment from reviewer 2022-02-25:

Thanks for the explanation. I will kindly suggest to report n as number of subjects and k number of treatments so to have an understandable link between the method description and the results.

  1. Control of the testing materials alone is missing. As such, the conclusions based on OD measurements need to be revised in the light of background levels.

Answer) Fig 3B to D, we expressed the intensity of special staining as relative OD, which means absolute OD of backgrounds were converted into 1 and the measured ODs were expressed as relative OD compare with backgrounds.

What is the OD (at 490nm) of the ASD alone at different concentrations reported in fig 1?

New comments: This question remains unanswered

Answer) For crystal violet, safranin, concanavalin A staining, we remove culture media and washed with PBS, and then stained. During removing media, ASD in medium also be removed. So ASD did not influence the OD value. To clarify ‘, after removing the culture media’ was added in line 293 as line 284.

New comment from reviewer 2022-02-25:

I will kindly suggest to the authors to check the OD of different concentrations of ASD without the cell so to check the background levels associated to ASD alone. This is critically relevant in the figure 1.

Author Response

I thank the editors and referees of the ‘International Journal of Molecular Sciences’ for taking their time to review my article.

I have made some corrections and tried to answer or explain about the questions.

New comment from reviewer 2022-02-25:

Thanks for the explanation. I will kindly suggest to report n as number of subjects and k number of treatments so to have an understandable link between the method description and the results.

 Answer) N number in Fig 1, 2, 3, and 4 was changed as k numbers, as recommended.

New comment from reviewer 2022-02-25:

I will kindly suggest to the authors to check the OD of different concentrations of ASD without the cell so to check the background levels associated to ASD alone. This is critically relevant in the figure 1.

Answer) Fig 1 was changed to show the relative cell survival curve (A) and absolute OD of different concentrations of ASD without cells, as recommended.

I hope the revised manuscript will better meet the requirements of the ‘International Journal of Molecular Sciences’ for publication.

Thank you.

Round 3

Reviewer 1 Report

no further comments

This manuscript is a resubmission of an earlier submission. The following is a list of the peer review reports and author responses from that submission.

Round 1

Reviewer 1 Report

The manuscript aims to characterize the toxic effects of Asian dust (ASD) and Aspergillus fumigatus (ASP) on human primary nasal epithelial cells. The authors assessed cell viability, the levels of IL6, IL8, and TGF beta in the supernatant as well as the biofilm formation/ fungal extracellular matrix.

The authors concluded on the inflammatory potential of ASD and ASP and their ability to favor biofilm formation in vitro.

The manuscript is well organized and reads well however there is a number of issues that the manuscript suffers from.

  • It is unclear for the reviewer which type of plate was used for the culture and viability test.
  • The number of the primary cell used in the exposure is missing.
  • Is there any characterization of the size or the shape of particles in ASD?
  • The authors reported that they performed the experiments at least 5 times and thus by reporting n=5 means 5 experiments. It was a little bit strange to run 5 in vitro experiments without replicating. Any specific reason for that? The reviewer wonders if it is the number of replicates and not the number of independent experiments the authors meant.
  • Control of the testing materials alone is missing. As such, the conclusions based on OD measurements need to be revised in the light of background levels.
  • Another critical issue is that authors reported in figure S2 protease activity, but there is no description of how that was measured in the method section.
  • The scale bars in fig 2 A-C are missing
  • The reviewer wonders if the authors can present the limitations of the approach adopted.
  • Lines 102-105: the meaning of the sentence is not clear for the reviewer
  • Line 151: autoclaving eliminate growth but not microbiological effects
  • Line 184-185: Any reference for that.

Author Response

I thank the editors and referees of the ‘International Journal of Molecular Sciences’ for taking their time to review my article.

I have made some corrections and added some results in the manuscript after going over the referee’s comments.

Reviewer 1.

It is unclear for the reviewer which type of plate was used for the culture and viability test.

Answer) To clarify the condition of cell culture and viability test, in line 261 and 264 ‘Cell suspensions (1 × 106 cells/ml) were plated in 6-well culture plates and grown to ….37oC.
To determine the cytotoxic effects of ASD, 200 μl of nasal epithelial cells (1 × 105 cells/ml) were incubated in 96-well tissue culture plates with 0, 10….’ were added in Material and Methods.

The number of the primary cell used in the exposure is missing.

Answer) In line 263, cell number was described as ‘200 μl of nasal epithelial cells (1 × 105 cells/ml) were incubated….’

Is there any characterization of the size or the shape of particles in ASD?

Answer) In line 159-160, size and main components were described as ‘The median size of ASD paticle was 6.06 μm and the most abundant component was SiO (52.1%) (Table S1).’

The authors reported that they performed the experiments at least 5 times and thus by reporting n=5 means 5 experiments. It was a little bit strange to run 5 in vitro experiments without replicating. Any specific reason for that? The reviewer wonders if it is the number of replicates and not the number of independent experiments the authors meant.

Answer) To clarify, authors described as ‘All the experiments were performed on at least five independent individuals and every experiment was performed in duplicate.’ In line 309-310.

Control of the testing materials alone is missing. As such, the conclusions based on OD measurements need to be revised in the light of background levels.

Answer) Fig 3B to D, we expressed the intensity of special staining as relative OD, which means absolute OD of backgrounds were converted into 1 and the measured ODs were expressed as relative OD compare with backgrounds.

Another critical issue is that authors reported in figure S2 protease activity, but there is no description of how that was measured in the method section.

Answer) The method measuring protease activity was added in the Supplement ‘Methods S3. Determination of protease activity. The protease activity of A. fumigatus was determined using a protease activity assay kit (Cayman, Ann Arbor, MI, USA). When nasal epithelial cells reached confluence, cells were incubated with A. fumigatus conidia (1 × 105/ml) for 2 h and then washed to remove the non-adherent fungi. Then, nasal epithelial cells were cultured in media, containing 50 µg/ml or 100 µg/ml ASD for 24 h, 48 h, and 72 h. 100 μl of cell-free supernatants were placed in 96-well black plates with 100 μl of protease substrate for 20 min at RT. The value of protease activity was determined with an excitation wavelength of 485 nm and an emission wavelength of 520 nm using FLUOstar Optima (BMG Labtech. Ortenaukreis, Germany).’

The scale bars in fig 2 A-C are missing

Answer) Scale bars were added in Fig 4 A-C.

The reviewer wonders if the authors can present the limitations of the approach adopted.

Answer) The limitation of this study was added in the last part of Discussion, line 227 to 234, as ‘There are some limitations in explaining the pathophysiologic phenomenon occurring in the sinonasal mucosa. First, ……… Second, …………….. Third, …….. conidia was used to develop biofilm, however we do not know the concentration or the number of conidia in the sinonasal mucosa.’

Lines 102-105: the meaning of the sentence is not clear for the reviewer

Answer) To clarify the expression of sentence, some words were added or changed as ‘Crystal violet has been used to determine extracellular matrix (ECM), cell viability and to quantify biofilm biomass [15]. When A. fumigatus was cultured with nasal epithelial cells, the quantity of ECM increased significantly at each time point comparison to the wells, which contained A. fumigatus conidia cultured without nasal epithelial cells. However, the coculture of ASD with A. fumigatus did not have a significantly increased crystal violet staining intensities in comparison with A. fumigatus incubated without ASD.’

Line 151: autoclaving eliminate growth but not microbiological effects

Answer) Microbiological effect was changed as ‘microbial effect….’

Line 184-185: Any reference for that.

Answer) Reference 22 was added in line 193.

 I hope the revised manuscript will better meet the requirements of the ‘International Journal of Molecular Sciences’ for publication.

Thank you.

Reviewer 2 Report

This manuscript intends to understand if the coexistence of ASD and A. fumigatus will increase the development of fungal biofilms and fungal inflammatory diseases in the nasal mucosa.

The manuscript seems to be well written and the results are interesting. Therefore, I would recommend the publication of this article in International Journal of Molecular Sciences. However, I have a few comments that I believe would improve the manuscript and thus need to be addressed.

I think the introduction needs to better reflect the objectives of the work. Authors should give a state-of-art on the effect of the ASD presence, along with A. fumigatus, on the nasal mucosa infection. They talk about these two factors independently, but no evidence is given on the merged effect and in what they had based to study this enhanced result.

In figure 1, what can explain the slight increase in all conditions on 100 ug/ml? The cell viability is higher compared with lower concentrations of ASD or even in the absence of ASD. What would be the explanation?

Figure 2: Formatting images is needed (mostly in third graphic)

Why not test the microbiological component of ASD? The biofilm formation could be highly dependent on the ASD microbial content. The authors in line 149, say that “Microbial material in ASD enhances allergic and infectious airway inflammation”. This would be worth to be tested or at least referenced. Also, in line 200-202 they clearly stated that “Bacterial and fungal biofilms coexist in CRS patients and these organisms synergistically interact to contribute to the development and survival of biofilms”.

Confocal images- in the printable scale it’s not possible to distinguish the different staining. The concanavalin A would not also stain the epithelial cells? In that matter, it is not possible to distinguish the epithelial cells in the images. The conditions ASD50+ASP and ASD100+ASP were not performed on epithelial cells? This is not clear. What is the reason for not performing in the cellular line?

Line 233 – Sampling time period is missing and how was the samples stored before experiments.

Author Response

I thank the editors and referees of the ‘International Journal of Molecular Sciences’ for taking their time to review my article.

I have made some corrections and added some results in the manuscript after going over the referee’s comments.

Reviewer 2.

I think the introduction needs to better reflect the objectives of the work. Authors should give a state-of-art on the effect of the ASD presence, along with A. fumigatus, on the nasal mucosa infection. They talk about these two factors independently, but no evidence is given on the merged effect and in what they had based to study this enhanced result.

Answer) The last part of Introduction, line 57-60, was revised to clarify the background and emphasize the purpose of this study as ‘ASD and A. fumigatus are risk factors for FB and CRS. The presence of ASD decreases host immune defense and increases cell susceptibility to bacterial infection and biofilm formation in airway mucosa [14]. ASD particles may provide a favorable surface for fun-gal attachment and biofilm growth.,

In figure 1, what can explain the slight increase in all conditions on 100 ug/ml? The cell viability is higher compared with lower concentrations of ASD or even in the absence of ASD. What would be the explanation?

Answer) The cell viability at 100 ug/ml of ASD tended to increase, this increase may be associated with the increased production of chemical mediators by 100 ug/ml of ASD than lower concentration of ASD in nasal epithelial cell (Fig 2).
The viability study was performed to determine the optimal condition for this study, so we did not mention about the increased viability in Discussion.
If we need to add, we will mention about that in Discussion.

Figure 2: Formatting images is needed (mostly in third graphic)

Answer) We changed Fig 2 as recommended.

Why not test the microbiological component of ASD? The biofilm formation could be highly dependent on the ASD microbial content. The authors in line 149, say that “Microbial material in ASD enhances allergic and infectious airway inflammation”. This would be worth to be tested or at least referenced. Also, in line 200-202 they clearly stated that “Bacterial and fungal biofilms coexist in CRS patients and these organisms synergistically interact to contribute to the development and survival of biofilms”.

Answer) Microbial components, like bacteria, fungus, may strongly influence the immune defense system of airway mucosa. The composition of microbial components may vary depending on the collected place, time, and season, but the composition variance of ASD particles may not be relatively large. So we used ASD particles for this study. We mentioned about that in line 154-157 as ‘The composition of the ASD mixture, that contain organic, inorganic, and microbial components, depends on the location where it is collected, the sources of emission near the collection site, and the season in which the dust is sampled.’

Confocal images- in the printable scale it’s not possible to distinguish the different staining. The concanavalin A would not also stain the epithelial cells? In that matter, it is not possible to distinguish the epithelial cells in the images. The conditions ASD50+ASP and ASD100+ASP were not performed on epithelial cells? This is not clear. What is the reason for not performing in the cellular line?

Answer) In order to increase the resolution of the picture, the picture in Figure 4 A-C were replaced. The experiment was conducted by culturing nasal epithelial cells and A. fumigatus together and then staining. Fig 4 A-C could not show the epithelial cells due to the confocal microscopic views were focused on fungi.

Line 233 – Sampling time period is missing and how was the samples stored before experiments.

Answer) To clarify, we added the processes for primary culture to line 257-259, as Specimens were placed in Ham’s F-12 medium supplemented with 100 IU of penicillin, 100 µg/ml streptomycin, and 2 µg/ml amphotericin B, and were then transported to the laboratory.,

I hope the revised manuscript will better meet the requirements of the ‘International Journal of Molecular Sciences’ for publication.

Thank you.

Round 2

Reviewer 1 Report

Round II

  1. Regarding the plates (6 wells and 96 wells), can the authors provide the product reference and producer? The reviewer considers this info as critical for other researchers that will try to repeat the experiment.

  1. The authors reported that they performed the experiments at least 5 times and thus by reporting n=5 means 5 experiments. It was a little bit strange to run 5 in vitro experiments without replicating. Any specific reason for that? The reviewer wonders if it is the number of replicates and not the number of independent experiments the authors meant.

Answer) To clarify, authors described as ‘All the experiments were performed on at least five independent individuals and every experiment was performed in duplicate.’ In line 309-310.

The authors run at least 5 experiments and every experiment in duplicate. The n for such setup will be 10. The reviewer is not convinced by the answer provided above and will appreciate if the authors can give an explanation for the n values in fig 1 (n=5), fig 2 and 3 (n=7), fig 4 (n=5.1), figS2 and figS4 (n=5).

  1. Control of the testing materials alone is missing. As such, the conclusions based on OD measurements need to be revised in the light of background levels.

Answer) Fig 3B to D, we expressed the intensity of special staining as relative OD, which means absolute OD of backgrounds were converted into 1 and the measured ODs were expressed as relative OD compare with backgrounds.

What is the OD (at 490nm) of the ASD alone at different concentrations reported in fig 1? this is critically important to uncover any interference from the ASD in the OD measurement.